## [Peer Review File · Royal Society Open Science]

Review History

RSOS-211309.R0 (Original submission)

Review form: Reviewer 1

Is the manuscript scientifically sound in its present form?

No

Are the interpretations and conclusions justified by the results?

Yes

Is the language acceptable?

Yes

Do you have any ethical concerns with this paper?

No

Have you any concerns about statistical analyses in this paper?

Yes

Recommendation?

Major revision is needed (please make suggestions in comments)

Comments to the Author(s)

This paper presents a very interesting experiment where dispersal among zooplankton mesocosms have been manipulated to test for effects on diversity and stability at the intra and interspecific levels. The experiment looks great, and the results are clear and match with theoretical predictions. I thus very much liked reading this manuscript. Some writing and analysis aspects would however need revisions:

- Why focussing the paper on habitat isolation, since what has been actually manipulated is dispersal rate? Habitat isolation consequences for movements might often be more complex than just decreasing dispersal rate. The results of this experiment clearly show effects of dispersal rate on diversity and stability in communities. That these results might inform about the potential consequences of habitat isolation is a matter of discussion. Focussing on the effects of dispersal on diversity and stability of communities would clearly not decrease the value of this paper, so I would suggest rewording the manuscript (basically only the first introduction paragraph, the discussion, and obviously title) to focus on dispersal and keep habitat isolation to the discussion. This would also help avoiding navigating between habitat isolation and dispersal rate when talking about the experiment (e.g. L. 164-165; 169 & 171... see also a comment below). Note that in such a case, some more explanation about the value of focussing on intra and interspecific levels might be useful.

- To compare community dynamics with and without dispersal, you indeed need to calculate community metrics at similar scales, so to make global habitat size equal at the scale of the metric. However, calculating metrics for only one tank per metacommunity seems weird to me, especially since you didn't explain how you choose which tank to use. If you want to keep with this method, you need to bootstrap your analyses for the choice of tank. But to my opinion, the better way of dealing with this data is to compute metrics at the scale of the metacommunity, so the mean or sum of all three tanks. This means that in the 0% dispersal treatment, intra and interspecific diversity and stability should be calculated for groups of 3 tanks. Since the experiment has not been performed this way, maybe bootstrap sampling for the 0% dispersal treatment among the 4 available tanks would do the job.

- L. 89: What is the diversity within *D. Pulex*? Are all clones different using the microsatellite markers?

- L. 82: How did you collect phytoplankton without any zooplankton? Please provide some details here.

- L. 102: Did you test for the efficiency of 60µm filtering to remove phytoplankton? Even not mentioning microalgae species that are >60µm, many smaller phytoplankton might stay in these filtered samples. I guess the proportion of phyto is likely to be small, but a basic quantification of whether dispersal only involves zooplankton or also (some?) phyto would be helpful.

- L. 111: please provide more explanation about how abundances of species were quantified (e.g., volume screened, identification method...)

- L. 115: what is the approximate density of *D. pulex*? This is important to point out here, to have an idea of how much 48 individuals are representative of this species population.

- L. 164: "increasing isolation and reduced dispersal capacity". You didn't manipulated both isolation and dispersal, only dispersal. Moreover, you didn't changed dispersal capacity, but rate of dispersal among communities. Please clarify this wording throughout the manuscript.

- Fig 1 & 3: Since the aim of these figures is to compare dynamics along dispersal rate, the figures would be easier to read with for instance one pannel per species/clone, each pannel presenting abundances for the 4 dispersal treatments.

Review form: Reviewer 2

Is the manuscript scientifically sound in its present form?

Yes

Are the interpretations and conclusions justified by the results?

Yes

Is the language acceptable?

Yes

Do you have any ethical concerns with this paper?

No

Have you any concerns about statistical analyses in this paper?

No

Recommendation?

Accept with minor revision (please list in comments)

Comments to the Author(s)

This paper presents the results of a metacommunity experiment with dispersal rate as manipulated variable, and with diversity and stability (temporal variability) as response variables. Diversity and stability are studied both at the interspecific and the intraspecific level. The experimental relationships with dispersal are similar at the two levels.

I find this study well conducted and interesting, and the paper well written. I have only a few minor comments.

l93: "20 individuals of each zooplankton taxon/clone were added to each mesocosm": This sentence is not clear about the number of individuals per *Daphnia pulex* clone: were there 20 or only 1? I assume there was only 1 individual, otherwise there would be 20 times more *D pulex* individuals than individuals of the other 5 species.

l101: "a source tank": I think this should be explained more explicitly. For a single dispersal event, was only 1 of the 3 tanks the source tank, or were the 3 tanks consecutively the source tank? I assume the latter, because on l105 dispersal is stated to be symmetric.

l111: "water removed for replacements was used as a plankton sample": From the response to comment 3 of the editor of *Biology Letters*, I understand that only 1 of the 3 tanks per metacommunity were sampled. Was this the same tank throughout the experiment?

l133: "we analyzed temporal stability using all sample dates": Did you also analyze temporal stability for the transitory dynamics and for the steady-state dynamics separately? Were the results similar?

l147: "We considered models that produced significant ($p < 0.05$) estimates of Asym and z as statistically significant": Do you mean by "significant estimates of Asym and z" that the estimates are significantly different from zero? This would be a weird for the estimate of z, because even for $z=0$ the function models an exponential decay towards Asym, namely with rate constant $\exp(z)=1$.

l206: "effects of dispersal were strongest for species whose abundances were most benefited by dispersal": This could be due to the fact that temporal variability quantified as $CV = \text{standard_deviation}/\text{mean}$ depends on the mean. It would be interesting to see whether the standard deviation shows the same trend.

l206: "as the zooplankton species used in our study are found in the same ecological guild of filter feeders, interspecific competition likely played a major role": This sentence is unclear to me.

l218: "the observed emigration rate was 0.009 per day": Was this observation done at a similar spatial scale as the 300 litres mesocosms of the experiment? If not, the emigration rates would not be comparable.

l233: "clone 29": This is no clone 29 in Figure 3; does this refer to the line labelled clone 28?

Decision letter (RSOS-211309.R0)

Dear Dr Steiner

The Editors assigned to your paper RSOS-211309 "Habitat isolation reduces intra- and interspecific biodiversity and stability." have now received comments from reviewers and would like you to revise the paper in accordance with the reviewer comments and any comments from the Editors. Please note this decision does not guarantee eventual acceptance.

Please submit your revised manuscript and required files (see below) no later than 21 days from today's (ie 15-Oct-2021) date. Note: the ScholarOne system will 'lock' if submission of the revision is attempted 21 or more days after the deadline. If you do not think you will be able to meet this deadline please contact the editorial office immediately.

on behalf of Dr Christie Bahlai (Associate Editor) and Pete Smith (Subject Editor)
 openscience@royalsociety.org

Associate Editor Comments to Author (Dr Christie Bahlai):

Comments to the Author:

We have now received two reviews of "Habitat isolation reduces intra- and interspecific biodiversity and stability." Both reviewers agreed that the manuscript was clearly written and examines an important subject, however Reviewer 1 had some important concerns about the framing of the paper as one about habitat isolation when they felt the focus of the experiment was dispersal rates. They also had actionable advice for alternate approaches to analysis. Reviewer 2 had some additional points that should be revised for clarity. Overall, I feel that this is a solid paper that requires some moderate revision to address some framing and clarity issues.

Reviewer comments to Author:

Reviewer: 1

Comments to the Author(s)

This paper present a very interesting experiment where dispersal among zooplankton mesocosms have been manipulated to test for effects on diversity and stability at the intra and interspecific levels. The experiment looks great, and the results are clear and match with theoretical predictions. I thus very much liked reading this manuscript. Some writing and analysis aspects would however need revisions:

- Why focussing the paper on habitat isolation, since what has been actually manipulated is dispersal rate? Habitat isolation consequences for movements might often be more complex than just decreasing dispersal rate. The results of this experiment clearly show effects of dispersal rate on diversity and stability in communities. That these results might inform about the potential consequences of habitat isolation is a matter of discussion. Focussing on the effects of dispersal on diversity and stability of communities would clearly not decrease the value of this paper, so I would suggest rewording the manuscript (basically only the first introduction paragraph, the discussion, and obviously title) to focus on dispersal and keep habitat isolation to the discussion. This would also help avoiding navigating between habitat isolation and dispersal rate when talking about the experiment (e.g. L. 164-165; 169 & 171... see also a comment below). Note that in such a case, some more explanation about the value of focussing on intra and interspecific levels might be useful.

- To compare community dynamics with and without dispersal, you indeed need to calculate community metrics at similar scales, so to make global habitat size equal at the scale of the metric. However, calculating metrics for only one tank per metacommunity seems wried to me, especially since you didn't explained how you choose which tank to use. If you want to keep with this method, you need to bootstrap your analyses for the choice of tank. But to my opinion, the better way of dealing with this data is to compute metrics at the scale of the metacommunity, so the mean or sum of all three tanks. This means that in the 0% dispersal tratment, intra and interspecific diversity and stability should be calculated for groups of 3 tanks. Since the experiment has not been performed this way, maybe bootstrap sampling for the 0% dispersal treatment among the 4 available tanks would do the job.

- L. 89: What is the diversity within *D. Pulex*? Are all clones different using the microsatellite markers?

- L. 82: How did you collected phytoplankton without any zooplankton? Please provide some details here.

- L. 102: Did you tested for the efficiency of 60um filtering to remove phytoplankton? Even not mentioning microalgae species that are >60um, many smaller phytoplankton might stay in these filtered samples. I guess the proportion of phyto is likely to be small, but a basic quantification of whether dispersal only involve zooplankton or also (some?) phyto would be helpful.

- L. 111: please provide more explanation about how abundances of species were quantified (e.g., volume screened, identification method...)

- L. 115: what is the approximate density of *D. pulex*? This is important to point out here, to have an idea of how much 48 individuals are representative of this species population.

- L. 164: "increasing isolation and reduced dispersal capacity". You didn't manipulated both isolation and dispersal, only dispersal. Moreover, you didn't changed dispersal capacity, but rate of dispersal among communities. Please clarify this wording throughout the manuscript.

- Fig 1 & 3: Since the aim of these figures is to compare dynamics along dispersal rate, the figures would be easier to read with for instance one pannel per species/clone, each pannel presenting abundances for the 4 dispersal treatments.

Reviewer: 2

Comments to the Author(s)

This paper presents the results of a metacommunity experiment with dispersal rate as manipulated variable, and with diversity and stability (temporal variability) as response variables. Diversity and stability are studied both at the interspecific and the intraspecific level. The experimental relationships with dispersal are similar at the two levels.

I find this study well conducted and interesting, and the paper well written. I have only a few minor comments.

l93: "20 individuals of each zooplankton taxon/clone were added to each mesocosm": This sentence is not clear about the number of individuals per *Daphnia pulex* clone: were there 20 or only 1? I assume there was only 1 individual, otherwise there would be 20 times more *D. pulex* individuals than individuals of the other 5 species.

l101: "a source tank": I think this should be explained more explicitly. For a single dispersal event, was only 1 of the 3 tanks the source tank, or were the 3 tanks consecutively the source tank? I assume the latter, because on l105 dispersal is stated to be symmetric.

l111: "water removed for replacements was used as a plankton sample": From the response to comment 3 of the editor of *Biology Letters*, I understand that only 1 of the 3 tanks per metacommunity were sampled. Was this the same tank throughout the experiment?

l133: "we analyzed temporal stability using all sample dates": Did you also analyze temporal stability for the transitory dynamics and for the steady-state dynamics separately? Were the results similar?

l147: "We considered models that produced significant ($p < 0.05$) estimates of $Asym$ and z as statistically significant": Do you mean by "significant estimates of $Asym$ and z " that the estimates are significantly different from zero? This would be a weird for the estimate of z , because even

for $z=0$ the function models an exponential decay towards Asym, namely with rate constant $\exp(z)=1$.

l196: "effects of dispersal were strongest for species whose abundances were most benefited by dispersal": This could be due to the fact that temporal variability quantified as $CV = \text{standard_deviation}/\text{mean}$ depends on the mean. It would be interesting to see whether the standard deviation shows the same trend.

l206: "as the zooplankton species used in our study are found in the same ecological guild of filter feeders, interspecific competition likely played a major role": This sentence is unclear to me.

l218: "the observed emigration rate was 0.009 per day": Was this observation done at a similar spatial scale as the 300 litres mesocosms of the experiment? If not, the emigration rates would not be comparable.

l233: "clone 29": This is no clone 29 in Figure 3; does this refer to the line labelled clone 28?

===PREPARING YOUR MANUSCRIPT===

===PREPARING YOUR REVISION IN SCHOLARONE===

Author's Response to Decision Letter for (RSOS-211309.R0)

See Appendix A.

Decision letter (RSOS-211309.R1)

Dear Dr Steiner

The Editors assigned to your paper RSOS-211309.R1 "Habitat isolation reduces intra- and interspecific biodiversity and stability." have now received comments from reviewers and would like you to revise the paper in accordance with the reviewer comments and any comments from the Editors. Please note this decision does not guarantee eventual acceptance.

Please note that it is unusual to be granted a further opportunity to substantially revise your paper; if you are unable to satisfy the Editors (or reviewers, if their guidance is sought following revision) that the paper is ready for acceptance, you may be rejected from further consideration, so please carefully amend your paper to respond to the Editor's concerns and provide point-by-point responses and rebuttals for the Editor to review.

Please submit your revised manuscript and required files (see below) no later than 21 days from today's (ie 22-Nov-2021) date. Note: the ScholarOne system will 'lock' if submission of the revision is attempted 21 or more days after the deadline. If you do not think you will be able to meet this deadline please contact the editorial office immediately.

on behalf of Dr Christie Bahlai (Associate Editor) and Pete Smith (Subject Editor)
openscience@royalsociety.org

Associate Editor Comments to Author (Dr Christie Bahlai):
Associate Editor
Comments to the Author:

I have now reviewed the revised copy of this manuscript and I feel it still has a ways to go before it will be ready for publication.

The authors did not take on board several of the points of feedbacks that the reviewers noted, and after a careful review of the revised document, I feel that there are still numerous points of clarity that should be resolved. Furthermore, I feel the results are under-synthesized- the data are presented in multiple raw panels of observations- it's really hard for a reader to glean any generalizations from the work. What is being tested? What 'should' the data look like? It's clear that the data were collected thoroughly and thoughtfully but the hows and whys of the analysis approach and what it all means when it's brought together isn't very clear in the current presentation.

For example, the four figures are essentially 16 individual figures, labeled as panels but they're not really brought together in a way that makes it meaningful to compare them. What's the take-home point for Figure 1 a-d? And the rest of the figures?

All the figures use a problematic colour scheme which makes it difficult for me to see patterns presented. The error bars also obscure the data quite dramatically. I would suggest re-constructing the figures in ggplot with a professionally-vetted color scheme (for example, colorbrewer has some nice, high contrast schemes for categorical data:

<https://colorbrewer2.org/#type=qualitative&scheme=Dark2&n=6>) and then layering the error as transparent ribbons in the background, and plotting the lines and points on top of those.

In addition, I found the response to review very dismissive of several reviewer concerns that I think require a bit more thought, especially on the line of thought about whether the manipulations genuinely represent habitat isolation.

Some specific line items:

21 "Field experiment" – this is more a semi- field mesocosm study, not an open system as 'field experiment' implies. Some of the reviewer comments about comparing dispersal rates to field rates might stem from this. This statement also appears in L 68, so I suggest revising to emphasize the mesocosm nature of the study.

122 "Molecular methods followed..." can you offer a very brief summary? This vague and would be difficult to reproduce in this form.

131- Please make the analysis code available in either an online repository or as a supplement

137- Biases exist in all data, but randomization should minimize it, revise to reflect this

138- Why use multiple diversity indices? This isn't wrong but needs to be explained- what is each index for?

145- which theory? Citation/specification needed

153- proportion of what? Please specify

158- I'd like to see this model more clearly specified- x, y and z are defined mathematically but not biologically- what variables do they represent?

===PREPARING YOUR MANUSCRIPT===

If you have been asked to revise the written English in your submission as a condition of publication, you must do so, and you are expected to provide evidence that you have received language editing support. The journal would prefer that you use a professional language editing service and provide a certificate of editing, but a signed letter from a colleague who is a fluent speaker of English is acceptable. Note the journal has arranged a number of discounts for authors using professional language editing services (<https://royalsociety.org/journals/authors/benefits/language-editing/>).

===PREPARING YOUR REVISION IN SCHOLARONE===

-- Ensure that your data access statement meets the requirements at <https://royalsociety.org/journals/authors/author-guidelines/#data>. You should ensure that you cite the dataset in your reference list. If you have deposited data etc in the Dryad repository, please include both the 'For publication' link and 'For review' link at this stage.

Author's Response to Decision Letter for (RSOS-211309.R1)

See Appendix B.

Decision letter (RSOS-211309.R2)

Dear Dr Steiner,

It is a pleasure to accept your manuscript entitled "Habitat isolation reduces intra- and interspecific biodiversity and stability." in its current form for publication in Royal Society Open Science. The comments of the reviewer(s) who reviewed your manuscript are included at the foot of this letter.

on behalf of Dr Christie Bahlai (Associate Editor) and Pete Smith (Subject Editor)
openscience@royalsociety.org

Appendix A

COLLEGE OF LIBERAL ARTS
AND SCIENCES

CHRISTOPHER F. STEINER
ASSOCIATE PROFESSOR
DEPARTMENT OF BIOLOGICAL SCIENCES
WAYNE STATE UNIVERSITY
5047 GULLEN MALL, #3120
DETROIT, MI 48202
(313) 577-0728 OFFICE
EMAIL: csteiner@wayne.edu

November 12, 2021

To whom it may concern:

Thank you for the opportunity to revise our manuscript: "Habitat isolation reduces intra- and interspecific biodiversity and stability".

Below we address the reviewers' comments and describe the changes we have made to our paper.

Thank you for considering our manuscript.

Sincerely,

Christopher Steiner

Reviewer #1 comments:

- Why focossing the paper on habitat isolation, since what has been actually manipulated is dispersal rate? Habitat isolation consequences for movements might often be more complex than just decreasing dispersal rate. The results of this experiment clearly show effects of dispersal rate on diversity and stability in communities. That these results might inform about the potential consequences of habitat isolation is a matter of discussion. Focussing on the effects of dispersal on diversity and stability of communities would clearly not decrease the value of this paper, so I would suggest rewording the manuscript (basically only the first introduction paragraph, the discussion, and obiously title) to focus on dispersal and keep habitat isolation to the discussion.

We disagree somewhat with the reviewer. In the literature, including classic metacommunity models, increasing habitat isolation is commonly equated with decreasing immigration rates. We agree that isolation in the wild can involve many underlying mechanisms which will impact dispersal rates. We now acknowledge this in our paper (lines 267-275). We point out that our experiment is a simplified representation of the effect of isolation and now mention some of the complexities of isolation in nature.

- To compare community dynamics with and without dispersal, you indeed need to calculate community metrics at similar scales, so to make global habitat size equal at the scale of the metric. However, calculating metrics for only one tank per metacommunity seems wried to me, especially since you didn't explained how you choose which tank to use. If you want to keep with this method, you need to bootstrap your analyses for the choice of tank.

We now explain how we chose target tanks within each metacommunity (lines 130-133). As the metacommunities and single tank treatments were randomized within the field (lines 101-102), our methodology should provide an unbiased estimate of treatment effects. This type of target-neighbor approach for analyzing metacommunity dynamics has been used previously (e.g. Steiner et al. 2011 *American Naturalist* 178: 159-170 ; Steiner et al. 2016 *Proceedings of the Royal Society of London B: Biological Sciences* 283: 20152426). We do not understand how choice of tanks can be “bootstrapped”

the better way of dealing with this data is to compute metrics at the scale of the metacommunity, so the mean or sum of all three tanks. This means that in the 0% dispersal treatment, intra and interspecific diversity and stability should be calculated for groups of 3 tanks. Since the experiment has not been performed this way, maybe bootstrap sampling for the 0% dispersal treatment among the 4 available tanks would do the job.

Sampling was only performed for target tanks within each metacommunity. Thus, we are unable to compute response variables across all three tanks. We do not understand why bootstrap resampling of any sort is required for our analyses. Our experimental design and analyses using standard parametric techniques are adequate and should provide robust, unbiased estimates of statistical significance.

- L. 89: *What is the diversity within D. Pulex? Are all clones different using the microsatellite markers?*

As stated in our methods, the clones added to the tanks were all identifiable as unique multi-locus genotypes using microsatellite markers (line 91). However, note that we caught a mistake in our methods. We used 10 clones (as shown in our figures), not 20 clones as originally stated (line 90).

- L. 82: *How did you collected phytoplankton without any zooplankton? Please provide some details here.*

Methods are now provided (lines 84-87)

- L. 102: *Did you tested for the efficiency of 60um filtering to remove phytoplankton? Even not mentionning microalgae species that are >60um, many smaller phytoplankton might stay in these filtered samples. I guess the proportion of phyto is likely to be small, but a basic quantification of whether dispersal only involve zooplankton or also (some?) phyto would be helpful.*

We did not quantify phytoplankton composition. However, we acknowledge that some movement of phytoplankton and bacteria likely occurred during dispersal events and mention this caveat in our Discussion (lines 197-201)

- L. 111: *please provide more explanation about how abundances of species were quantified (e.g., volume screened, identification method...)*

Now provided on lines 116-118.

- L. 115: *what is the approximate density of D. pulex? This is important to point out here, to have an idea of how much 48 individuals are representative of this species population.*

Mean densities for all treatments are displayed in figure 1. Raw data for all replicates will be provided to readers upon publication. We disagree with the reviewer that knowing “how much 48 individuals are representative” is vital. What’s more important in terms of having unbiased estimates of diversity is that sampling effort was equal across all samples, regardless of total densities. However, upon reviewing our data, we noted that some samples produced fewer individuals due to small population sizes and were inadvertently included in our analyses. To increase robustness, we have now removed these lower quality data points from our analyses of *D. pulex* intraspecific diversity and in associated figures (explained in lines 126-129). These changes did not greatly alter our statistical results or general conclusions (lines 252-255). Indeed, one statistical test that was borderline significant previously became statistically significant.

- L. 164: *"increasing isolation and reduced dispersal capacity". You didn't manipulated both isolation and dispersal, only dispersal. Moreover, you didn't changed dispersal capacity, but rate of dispersal among communities. Please clarify this wording throughout the manuscript.*

As noted above, we disagree this characterization of isolation. As explained above, many models equate increasing isolation with decreasing dispersal rate. This is also identified as the major consequence of habitat isolation in the literature dealing with habitat fragmentation. Dispersal capacity and dispersal rate are synonymous.

- Fig 1 & 3: *Since the aim of these figures is to compare dynamics along dispersal rate, the figures would be easier to read with for instance one pannel per species/clone, each pannel presenting abundances for the 4 dispersal treatments.*

We disagree. The primary intent of these figures is to see how diversity (inter- and intraspecific) change with dispersal rate. We have chosen to retain the original figures.

Reviewer #2 comments:

l93: *"20 individuals of each zooplankton taxon/clone were added to each mesocosm": This sentence is not clear about the number of individuals per Daphnia pulex clone: were there 20 or only 1? I assume there was only 1 individual, otherwise there would be 20 times more D pulex individuals than individuals of the other 5 species.*

We now clarify (lines 96-97)

l101: *"a source tank": I think this should be explained more explicitly. For a single dispersal event, was only 1 of the 3 tanks the source tank, or were the 3 tanks consecutively the source tank? I assume the latter, because on l105 dispersal is stated to be symmetric.*

We have changed some wording (line 106 and 110) to make this clearer. All tanks in a metacommunity were sampled and dispersal was reciprocal among them.

l111: "water removed for replacements was used as a plankton sample": From the response to comment 3 of the editor of Biology Letters, I understand that only 1 of the 3 tanks per metacommunity were sampled. Was this the same tank throughout the experiment?

We have made this clearer (lines 134-137). The same tank from each metacommunity was sampled throughout the experiment.

l133: "we analyzed temporal stability using all sample dates": Did you also analyze temporal stability for the transitory dynamics and for the steady-state dynamics separately? Were the results similar?

Viewing dynamics in figure 1, it doesn't appear that any "steady state" is attained. Rather, dynamics appear in flux throughout the experiment (with some taxa showing steady increases over time, other showing increases followed by decreases in density). It is not clear how these could be divided into "transitory" versus "steady state" phases.

l147: "We considered models that produced significant ($p < 0.05$) estimates of $Asym$ and z as statistically significant": Do you mean by "significant estimates of $Asym$ and z " that the estimates are significantly different from zero? This would be a weird for the estimate of z , because even for $z=0$ the function models an exponential decay towards $Asym$, namely with rate constant $exp(z)=1$.

We are grateful for the reviewer pointing this out. It's not just the z parameter that can equal zero, but all the parameters could equal zero. The main requirement for the regression to show a "significant" increasing or decreasing saturating pattern is for either the $Asym$ and y_0 parameters (or both) to be significantly different from zero. We have updated our methods accordingly (line 160).

l196: "effects of dispersal were strongest for species whose abundances were most benefited by dispersal": This could be due to the fact that temporal variability quantified as $CV = \text{standard_deviation}/\text{mean}$ depends on the mean. It would be interesting to see whether the standard deviation shows the same trend.

We disagree with this suggestion. A common issue is that variances (and standard deviations) scale with the mean. Using the CV removes this effect and allows comparisons of proportional variation around means. If you compared two treatments and they had equal standard deviations but different means, the treatment with the larger mean would have a lower CV (indicating greater stability). This makes sense; the comparison of CVs shows us rightly that proportional variation around the larger mean is lower compared to the treatment with the smaller mean. Both factors have to be considered (standard deviation and the mean) in order to elucidate this.

l206: "as the zooplankton species used in our study are found in the same ecological guild of filter feeders, interspecific competition likely played a major role": This sentence is unclear to me.

The term "ecological guild" is very common, indicating a group of species that utilize similar resources. Thus, such species potentially compete with each other. The zooplankton we used are all filter-feeders and thus in the same ecological guild (in contrast to some zooplankton species that are predators).

l218: "the observed emigration rate was 0.009 per day": Was this observation done at a similar spatial scale as the 300 lites mesocosms of the experiment? If not, the emigration rates would not be comparable.

We state in this sentence that this was observed in nature, in natural ponds. It seems obvious that natural waterbodies would not be same volume as a small experimental tank. We provide this as it's the only reported zooplankton dispersal rate that we know of that was measured in nature. It provides some context to the rates used in our study.

l233: "clone 29": This is no clone 29 in Figure 3; does this refer to the line labelled clone 28?

We have corrected figure 3, changing clone "28" to "29".

Appendix B

COLLEGE OF LIBERAL ARTS
AND SCIENCES

CHRISTOPHER F. STEINER
ASSOCIATE PROFESSOR
DEPARTMENT OF BIOLOGICAL SCIENCES
WAYNE STATE UNIVERSITY
5047 GULLEN MALL, #3120
DETROIT, MI 48202
(313) 577-0728 OFFICE
EMAIL: csteiner@wayne.edu

January 3, 2022

To whom it may concern:

Thank you for the opportunity to revise our manuscript: "Habitat isolation reduces intra- and interspecific biodiversity and stability".

Below we address the reviewer's comments (summarized in italics) and describe the changes we have made to our paper.

Thank you for considering our manuscript.

Sincerely,

Christopher Steiner

the data are presented in multiple raw panels of observations- it's really hard for a reader to glean any generalizations from the work. What is being tested? What 'should' the data look like?...For example, the four figures are essentially 16 individual figures, labeled as panels but they're not really brought together in a way that makes it meaningful to compare them. What's the take-home point for Figure 1 a-d? And the rest of the figures?

There were two basic predictions in our study: 1. increasing dispersal rate will increase diversity at intra- and interspecific levels and 2. increasing dispersal rate will reduce temporal variability (i.e. increase temporal stability) at intra- and interspecific levels. Figure 2 displays interspecific diversity and stability responses. Figure 4 displays intraspecific diversity and stability responses. The other two figures (Figure 1 and Figure 2) display dynamics of species and clones, respectively. Measures of diversity and stability were derived from these dynamic responses which is why we decided to display them in the main text. Figure 1 and 3 could be moved to the online supplement if it is felt that this will enhance clarity.

To better aid the reader, we have added letters to the panels in figures 1 and 3 and refer to specific panels in our explanation of results to guide the reader. We have added clarifying sentences throughout our descriptions of the results/figures, reiterating general predictions and clarifying how the figures relate to these predictions (e.g. lines 189-193, 216-218, 224-227, 263-269, 277-280).

All the figures use a problematic colour scheme which makes it difficult for me to see patterns presented. The error bars also obscure the data quite dramatically. I would suggest re-constructing the figures in ggplot with a professionally-vetted color scheme (for example, colorbrewer has some

nice, high contrast schemes for categorical data:

<https://colorbrewer2.org/#type=qualitative&scheme=Dark2&n=6>) and then layering the error as transparent ribbons in the background, and plotting the lines and points on top of those.

We have changed the color schemes in our figures as requested. We attempted to use error ribbons, as suggested by the reviewer, but found that for some figures this made the displays harder to decipher when compared to using bars. An example is provided below.

Thus, we have decided to display these figures using error bars (rather than ribbons). After consulting with the editors (Drs. Pete Smith and Andrew Dunn), both concurred with our decision. For figures 1 and 3, we use error bars without horizontal “hats” which we feel helps to make the figures easier to decipher.

I found the response to review very dismissive of several reviewer concerns that I think require a bit more thought, especially on the line of thought about whether the manipulations genuinely represent habitat isolation.

We have expanded on our discussion of the meaning of habitat isolation in our Discussion section (lines 300-305) and provide appropriate caveats. We note that isolation can have two inter-related effects: 1) it can reduce overall dispersal rates among patches (for passive dispersers like our study species, this may be driven by effects on animal vectors, such as increased mortality in the surrounding landscape and longer travel times among patches) and 2) changes in patterns of dispersal network structure and connectivity. We note that by utilizing manipulations in reciprocal dispersal rates among patches in our study, we only examined one aspect of habitat isolation, i.e., effects of changes in dispersal rate independent of changes in network topology (mechanism 1 independent of mechanism 2). We make clear in our Discussion (lines 302-305) that changes in dispersal network topology is a vital aspect of habitat isolation that we can't address and warrants further research in our study system. We have limited our discussion and review of this topic as we cannot address hypotheses with our study.

In our previous response to reviewers, we provided explanations for suggestions that we did not integrate into our manuscript.

21 *“Field experiment” – this is more a semi- field mesocosm study, not an open system as ‘field experiment’ implies. Some of the reviewer comments about comparing dispersal rates to field rates might stem from this. This statement also appears in L 68, so I suggest revising to emphasize the mesocosm nature of the study.*

We have added “mesocosm” as suggested

122 *“Molecular methods followed...” can you offer a very brief summary? This vague and would be difficult to reproduce in this form.*

Molecular methods are now provided in an online supplement

131- *Please make the analysis code available in either an online repository or as a supplement*

R code for statistical analyses are now provided in an online supplement

137- *Biases exist in all data, but randomization should minimize it, revise to reflect this*

We have revised the statement (lines 141-142)

138- *Why use multiple diversity indices? This isn't wrong but needs to be explained- what is each index for?*

We now provide brief explanation (lines 144-146).

145- *which theory? Citation/specification needed*

References are now provided

153- *proportion of what? Please specify*

We now specify “proportion of tank volume”.

158- *I'd like to see this model more clearly specified- x, y and z are defined mathematically but not biologically- what variables do they represent?*

We now describe what the model parameters refer to (lines 164-168)